# Evaluation of *Aedes aegypti*, *Aedes albopictus*, and *Culex quinquefasciatus* Mosquitoes Competence to *Oropouche virus* Infection

**DOI:** 10.3390/v13050755

**Published:** 2021-04-25

**Authors:** Silvana F. de Mendonça, Marcele N. Rocha, Flávia V. Ferreira, Thiago H. J. F Leite, Siad C. G. Amadou, Pedro H. F. Sucupira, João T. Marques, Alvaro G. A. Ferreira, Luciano A. Moreira

**Affiliations:** 1Mosquitos Vetores: Endossimbiontes e Interação Patógeno-Vetor, Instituto René Rachou—Fiocruz, Belo Horizonte 30190-002, MG, Brazil; smendonca@aluno.fiocruz.br (S.F.d.M.); marcelebio@yahoo.com.br (M.N.R.); psucupira@aluno.fiocruz.br (P.H.F.S.); alvaro.ferreira@fiocruz.br (A.G.A.F.); 2Departamento de Bioquímica e Imunologia, Instituto de Ciências Biológicas, Universidade Federal de Minas Gerais, 6627-Pampulha, Belo Horizonte 31270-901, MG, Brazil; fvianaferreira@gmail.com (F.V.F.); thjfl21@gmail.com (T.H.J.F.L.); gbadeguetchin@gmail.com (S.C.G.A.); jtmarques2009@gmail.com (J.T.M.); 3Faculté des Sciences de laVie, Université de Strasbourg, CNRS UPR9022, Inserm U1257, 67084 Strasbourg, France

**Keywords:** Oropouche, vector competence, urban epidemics, *Aedes aegypti*, *Aedes albopictus*, *Culex quinquefasciatus*

## Abstract

The emergence of new human viral pathogens and re-emergence of several diseases are of particular concern in the last decades. *Oropouche orthobunyavirus* (OROV) is an arbovirus endemic to South and Central America tropical regions, responsible to several epidemic events in the last decades. There is little information regarding the ability of OROV to be transmitted by urban/peri-urban mosquitoes, which has limited the predictability of the emergence of permanent urban transmission cycles. Here, we evaluated the ability of OROV to infect, replicate, and be transmitted by three anthropophilic and urban species of mosquitoes, *Aedes aegypti*, *Aedes albopictus*, and *Culex quinquefasciatus*. We show that OROV is able to infect and efficiently replicate when systemically injected in all three species tested, but not when orally ingested. Moreover, we find that, once OROV replication has occurred in the mosquito body, all three species were able to transmit the virus to immunocompromised mice during blood feeding. These data provide evidence that OROV is restricted by the midgut barrier of three major urban mosquito species, but, if this restriction is overcome, could be efficiently transmitted to vertebrate hosts. This poses a great risk for the emergence of permanent urban cycles and geographic expansion of OROV to other continents.

## 1. Introduction

The emergence of new human viral pathogens and re-emergence of several diseases have been of particular concern in the last decades [1,2,3]. Arboviruses (an acronym for “arthropod-borne viruses”) have become a major and constant threat to global health, with a high incidence of epidemic outbreaks in tropical and subtropical countries [4,5,6,7,8]. Although, several strategies to prevent, monitor, and contain arbovirus diseases continue to emerge and re-emerge in a manner that defies accurate predictions [4,9]. A recent example is the unexpected Zika pandemic in the Western Hemisphere, which has captured the attention, not only of public health professionals around the world, but also of researchers working in vector-borne infectious diseases [3,10,11]. With the likelihood that effective vaccines and clinically proven therapeutics are still many years out, arboviruses are an important and constant threat to human health worldwide, infecting millions of individuals and causing a large social and economic burden [12,13].

To be transmitted and maintained in nature, arboviruses require a vertebrate host (usually mammals or birds) and an arthropod vector, such as mosquitoes or biting midges, in which they must replicate prior to transmission [14,15,16]. All known arboviruses that infect humans have, or had in the past, sylvatic transmission cycles involving wild animals as reservoir hosts [14,16]. Most of these arboviruses infect humans “accidentally” via direct spillover from these sylvatic cycles [14]. Normally, humans do not develop sufficient viremia to maintain human–vector–human cycles and, therefore, are dead-end hosts [14,15]. However, some of the most important arboviruses for public health acquired the capacity of amplification in humans, bypassing the need for the sylvatic cycles, thus allowing the emergence of urban cycles [15,17]. Historically, the best example of arbovirus that evolved to undergo direct human amplification (human–mosquito–human) is dengue virus (DENV), a pathogen that currently infects around 100 million people each year worldwide and is found in over 100 countries [5,13,15,18]. Furthermore, arboviruses attracted interest in recent years owing to the unexpected emergence of chikungunya virus (CHIKV, 2013) and Zika virus (ZIKV, 2015) urban epidemics in the Americas [3,19,20,21]. Although many determinants of these two arboviruses’ emergence have an anthropological basis, the capacity of amplification in humans associated with a high incidence of the anthropophilic competent vectors such as *Aedes aegypti* and *Aedes albopictus* mosquitoes was of paramount importance for the urbanization of both CHIKV and ZIKV [4,22].

The recent emergence of CHIKV and ZIKV reinforces the need to identify novel arboviruses with potential for urbanization. With more than 100 arboviruses implicated in human diseases, the possibility of new arboviral urbanizations should not be underestimated [23]. Several factors are associated with arboviruses’ emergence and urbanization, such as development of global transportation and mobility, rising human population densities, and encroaching on wild habitats [4]. However, two fundamental aspects are required for arboviral urbanization. First, the arbovirus has to reach sufficient viremia in humans to extend the transmission cycle. Indeed, this requirement has been observed in urban outbreaks of Yellow fever virus (YFV) and ZIKV, with titers reaching 10^5–6^ and 10^3^ infectious units/mL, respectively [4]. Second, the presence of urban mosquito vectors exhibiting competence for the novel arbovirus infection, replication, and transmission, allowing the shift from the sylvatic vector to the urban vector [4]. The public health impact of the vector shifts are epitomized by the recent urban emergence of the ZIKV and CHIKV outbreaks in the Americas. The establishment of permanent urban outbreaks was facilitated by the fact that the viruses were able to jump from their original sylvatic vector into an urban vector species, such as *Aedes aegypti* and *Aedes albopictus*, where ZIKV and CHIKV were able to efficiently adapt [10,16,24]. Usually, these adaptations often rely on genetic mutations that increase the virus fitness in the new mosquito species, allowing a successful infection and replication as well as sustained transmission. The involvement of genetic mutations in arboviral emergence and urbanization can be exemplified by the emergence of CHIKV urban outbreaks, where small sequential adaptive mutations enhanced infection of *Aedes albopictus*, resulting in unprecedented epidemic activity and geographic expansion since 2004 [25,26].

In the past decades, several outbreaks involving sylvatic arboviruses have been causing urban epidemics involving thousands of people throughout tropical and subtropical regions [3]. An important example is *Oropouche orthobunyavirus* (OROV), an arbovirus endemic to South and Central America tropical regions, responsible for more than 30 epidemic events in the last six decades [27,28]. OROV causes an acute febrile illness (known as Oropouche Fever), lasting between two and seven days, and is typically accompanied by headache, myalgia, arthralgia, anorexia, dizziness, chills, and photophobia [28]. OROV is a member of the *Orthobunyavirus* genus in the Peribunyaviridae family and has a single-stranded, negative-sense RNA lipid-enveloped genome, divided into three segments (S, M, and L) protected by a nucleocapsid [16,27]. OROV is maintained in nature in sylvatic cycles involving wild mammals (sloths and non-human primates) as reservoir and amplification hosts, and mosquitoes such as *Coquillettidia venezuelensis*, *Aedes serratus*, *Culex quinquefasciatus*, and biting midges *Culicoides paraensis* [27]. However, several of the most important human outbreaks were associated with urban transmission cycles. These urban OROV epidemics were often connected with the vector *Culicoides paraensis* [27]. Additionally, it has been reported that mosquitoes *Cx*. *quinquefasciatus* can contribute to urban transmission of OROV, but to a lesser extent owing to the apparent low efficiency of the virus transmission by this mosquito species [29]. Despite several studies conducted to identify the putative urban hosts, no vertebrates other than humans have been implicated. However, currently available data on OROV urban transmission do not support the possibility of a human-to-human cycle maintained by anthropophilic mosquitoes. Although urban epidemics caused over half a million human infections since first identified, OROV urban outbreaks were self-limited [16,30,31,32,33]. This suggests that OROV is able to start outbreaks in humans, mainly in regions on the fringes of forested areas, that evolve into self-limited outbreaks that eventually die out. Despite being considered to have the potential to establish permanent urban transmission cycles, the lack of knowledge regarding the competence of urban/peri-urban mosquitoes to transmit OROV has limited the predictability of the emergence of permanent urban epidemics.

Here, we evaluated the ability of OROV to infect, replicate, and be transmitted by three anthropophilic and urban species of mosquitoes that are very abundant in urban areas throughout the Americas: *Ae*. *aegypti*, *Ae. Albopictus*, and *Cx*. *quinquefasciatus*. We have used three different routes to deliver the virus to the mosquitoes, infectious artificial blood meals, viremic mice, and systemic nano-injection, to show that these urban mosquitoes are susceptible to OROV infection, but this is restricted at the midgut level when the virus is delivered by the oral route. Nevertheless, if the virus reaches systemic infection by artificial injection, it is capable of replicating and being transmitted to a vertebrate host. These results raise concerns that OROV may be able to establish urban cycles.

## 2. Materials and Methods

### 2.1. Mosquito Lineages

Three mosquito species were used in this study, *Aedes aegypti*, *Aedes albopictus*, and *Culex quinquefasciatus*. For *Ae. aegypti*, we used three different population strains, the BH strain recently established (5 laboratory generations) from field-collected specimens in Belo Horizonte (Brazil); the RJ strain established (15 laboratory generations) from field-collected specimens in Rio de Janeiro (Brazil); and the laboratory strain, Bangkok (BKK). For *Ae. albopictus,* we used two different strains, the BH strain recently established (7 laboratory generations) from field-collected specimens in Belo Horizonte (Brazil), and the RJ strain established (12 laboratory generations) from field-collected specimens in Rio de Janeiro (Brazil). For *Cx. quinquefasciatus*, we used one population strain (BH strain) established (17 laboratory generations) from field-collected specimens in Belo Horizonte (Brazil).

### 2.2. Mice Lineages and OROV Inoculation

In this study, we used AG129 mice (IFN α/β/γ R^−/−^), a double knockout immunocompromised linage that lacks both types of interferon receptors, type I interferon (IFN α/β) and II IFN (IFN γ) [34]. AG129 mice were bred and maintained at the Animal Facility of the Instituto René Rachou, Fiocruz Minas. To infect mice with OROV, three-week-old AG129 mice were inoculated by intraperitoneal injection with 10^6^ p.f.u. mL^−1^ of OROV (0.1 mL of total volume injected). Experiments were approved by the Institutional Animal Care and Use Committee, Comissão de Ética no Uso de Animais da Fiocruz (CEUA) and performed according to institutional guidelines (license number LW-26-20).

### 2.3. Mosquito Rearing

All mosquito strains were reared under insectarium controlled conditions, 28 °C and 70–80% relative humidity, in a 12/12 h light/dark cycle. Eggs were placed in plastic trays containing two litres of filtered tap water supplemented with fish food (Tetramin, Tetra) for hatching and larvae were maintained at a density of 200 larvae per tray. After emerging, adults were kept in 30 × 30 × 30 cm BugDorm insect cages where mosquitoes were fed with 10% sucrose solution ad libitum.

### 2.4. Virus Propagation and Titration

In this study, we used the *Oropouche orthobunyavirus* strain BeAn19991 (prototype strain Brazilian isolated from the blood of a three-toed sloth, *Bradypus tridactylus*, in the Amazon region) with GenBank accession numbers KP052850, KP052852, and KP052851.1 [35,36]. OROV strain BeAn19991 was kindly supplied by Dr. Betânia Drumond from the Laboratório de Vírus, Universidade Federal de Minas Gerais, Brasil. OROV was propagated in Vero (African green monkey kidney) cells maintained on Dulbecco’s modified Eagle’s medium—high glucose (DMEM-High, Sigma Aldrich, MO, USA), supplemented with 5% FBS (fetal bovine serum), penicillin, and streptomycin. Briefly, cells were seeded to 70% confluency and infected at a multiplicity of infection (MOI) of 0.01. Cultures were maintained for 4 days at 37 °C, when supernatant was collected, cells were lysed by repeated freezing and thawing to release virus particles, and then mixed with the supernatant. After clarifying the supernatant by centrifugation, virus stocks were kept at −80 °C before use. Both OROV stocks and OROV from mice serum were titrated in Vero cells in six-well tissue culture plates. We allowed the virus to adsorb for 1 h at 37 °C, then an overlay of 2% in carboxymethyl cellulose (CMC) in DMEM with 2% FBS was added. Plates were incubated at 37 °C and 5% CO_2_ for 5 days. Then formaldehyde was added, and cells were covered with a crystal violet stain (70% water, 30% methanol, and 0.25% crystal violet) to visualize plaques.

### 2.5. Mosquito OROV Inoculations

For infections through membrane feeding, 5- to 6-day-old adult females were starved for 24 h and fed with a mixture of blood and virus supernatant containing 6 × 10^6^ p.f.u. mL^−1^ of OROV (0.5 mL of human blood and 1ml of virus supernatant) using a glass artificial feeding system covered with pig intestine membrane. Mosquitoes were allowed to feed for 30 min. After blood feeding, fully engorged females were selected and harvested individually for RNA extraction or dissection at different time points. Mosquitoes were anaesthetized with CO_2_ and kept on ice during the whole procedure. For mosquito infections on mice, we used OROV viremic AG129 mice, an immunodeficient murine animal model previously described for Flavivirus infections [37,38,39,40]. Three-week-old AG129 mice were inoculated by intraperitoneal injection with 10^6^ p.f.u. mL^−1^ of OROV. Infected mice were anaesthetized 3 or 4 d.p.i. (peak of viremia) using ketamine/xylazine (80/8 mg kg^−1^) and placed on top of the netting-covered containers with 5- to 6-day-old adult mosquito females. Mosquitoes were allowed to feed on mice for 1 h. After blood feeding, fully engorged females were selected and harvested individually for RNA extraction or dissection at different time points. To systemically infect mosquitoes, each individual female mosquito was injected with 1358 p.f.u. of OROV (70 nL using a Nanojet III, Drummond Scientific, Broomall, PA, USA), unless stated otherwise. Injection was performed in the paratergite region present in the mosquito thorax. After injection, recovered females were selected and harvested individually for RNA extraction or dissection at different time points.

### 2.6. OROV Transmission from Mosquitoes to Mice

To test whether mosquitoes were able to transmit OROV to AG129 mice, we first systemically inject female mosquitoes with 1358 p.f.u. of OROV (70 nL using a Nanojet III, Drummond Scientific, Broomall, PA, USA). Fourteen days after the OROV infection, female mosquitoes were allowed to feed in 3-week-old anaesthetised naive AG129 mice. Five female mosquitoes were exposed to each AG129 mice for 1 h and, after blood feeding, fully engorged females were selected and harvested individually for RNA extraction or dissection for OROV quantification. Three to four days after the AG129 mice were exposed to the mosquito bites, blood was collected for OROV quantification.

### 2.7. RNA Extraction and RT-qPCR

Tissues or whole mosquitoes were ground in TRIzol (Invitrogen) using glass beads, as previously described [41]. Total RNA was extracted from individual insects and subjected to quantitative real-time PCR (RT-qPCR) using the Power SYBR Green Master Mix (Applied Biosystems—Life Technologies, Foster City, CA, USA), following the manufacturer’s instructions. The viral RNA load was expressed relative to the endogenous control housekeeping gene, RPS17 for *Ae. aegypti*, RPL32 for *Ae. albopictus*, and 18s for *Cx. quinquefasciatus*. RPS 17 primers were Forward: 5′-TCC GTG GTA TCT CCA TCA AGC T-3′ and Reverse: 5′-CAC TTC CGG CAC GTA GTT GTC-3′. RPL32 primers were Forward: 5′-TAT GAC AAG CTT GCC CCC AA-3′ and Reverse: 5′-AGG AAC TTC TTG AAT CCG TTG G-3′. 18s primers were Forward: 5′-CGC GGT AAT TCC AGC TCC ACT A-3′ and Reverse: 5′-GCA TCA AGC GCC ACC ATA TAG G-3′. OROV primers were Forward: 5′-CAA CGA TGT ACC ACA ACG GAC TAC-3′ and Reverse: 5′-ACA ACA CCA GCA TTG AGC ACT T-3′.

### 2.8. Statistical Analysis

All statistical analyses were done using R, version 4.0.3, 10 October 2020 [42]. To compare the viral load between two groups we used a Mann–Whitney–Wilcoxon test in R. Multiple comparisons of viral loads were performed using a Kruskal–Wallis test in R.

### 2.9. Ethics Statement

The human blood used in all experiments was obtained from a blood bank (Fundação Hemominas), according to the terms of an agreement with the René Rachou Institute, Fiocruz/MG (OF.GPO/CCO agreement—Nr 224/16).

## 3. Results

### 3.1. Mosquitoes Ae. aegypti, Ae. albopictus, and Cx. quinquefasciatus Are Resistant to OROV Oral Infection after an Artificial Blood Meal

We first tested whether artificial blood meals containing OROV could be infectious to *Ae*. *aegypti*, *Ae. Albopictus*, and *Cx. quinquefasciatus* (Figure 1A, Appendix A). When *Ae*. *aegypti* females (5 to 7 days old) from a laboratory population—Bangkok strain (BKK)—were artificially fed on a mixture containing human blood and OROV at a final dose of 6.7 × 10^6^ p.f.u. mL^−1^, we were not able to detect OROV RNA in the mosquito abdomens, neither 7 days post feeding (d.p.f.) nor 14 d.p.f. (Figure 1B,C, respectively). To reduce putative effects associated with laboratory adaptation or genetic drift induced phenotype alterations, we also tested two different *Ae*. *aegypti* populations recently established (2 to 5 laboratory generations) from field-collected specimens, the BH strain from Belo Horizonte, Brazil, and the RJ strain from Rio de Janeiro, Brazil. In concordance with the Bangkok laboratory strain, we observed no OROV RNA in BH and RJ strains of *Ae*. *aegypti* mosquitoes artificially fed with OROV infectious blood (Figure 1B,C). This lack of OROV detection was consistent for both 7 and 14 d.p.f. (Figure 1B,C, respectively). To further investigate the role of other urban/peri-urban mosquito vectors, we also investigated the competence of *Ae. albopictus* and *Cx. quinquefasciatus* for OROV infection. For *Ae. albopictus*, we also used two different populations established from field-collected specimens, the BH strain from Belo Horizonte, Brazil, and the RJ strain from Rio de Janeiro, Brazil. For *Cx. quinquefasciatus*, we used a field-originated population from Belo Horizonte, Brazil, the BH strain. Corroborating our *Ae*. *aegypti*, we were not able to detect OROV RNA, neither in *Ae. albopictus* nor in *Cx. quinquefasciatus*, at both 7 and 14 d.p.f. (Figure 1B,C). Collectively, our results indicated that all three mosquito species are refractory to OROV oral infection.

### 3.2. Characterization of Oropouche virus (OROV) Infection in AG129 Mouse Model

To rule out any infectivity limitation due to an artificial blood meal and replicate the natural cycle of OROV transmission, which is vertebrate–mosquito–vertebrate, we first set out a murine model permissive for OROV infection. Although wild-type mice and other rodents (e.g., guinea pigs, hamsters, and rats) are resistant to most arboviral infection and disease, immunodeficient mouse models for viral infection have evolved with increasing success during the last two decades [43,44,45,46,47,48]. Mice that lack both the type I and II IFN receptors, AG129 mice (double Knockout for type I & II receptors -IFN-α/β and -γ), are now being used extensively in arbovirus studies [37,40,46,49,50,51,52]. Here, we inoculated juvenile AG129 mice with OROV through intraperitoneal (IP) injection, and analysed both survival rates and blood viremia kinetics (Figure 2A). We observed that the OROV infection was consistently lethal, with more than 80% (21/25) of the animals succumbing by day six post infection (p.i.) and 100% on day seven p.i. (Figure 2B). We also found that the animals start to succumb just five days post infection (Figure 2B). By contrast, no lethality was observed in the control group, where animals were injected only with mock solution (Figure 2B).

To evaluate the kinetics of blood viremia, we started by quantifying the viral load daily by RT-qPCR. We observed that, by two d.p.i., all animals already presented viral RNA in the blood (Figure 1C). After that, we found that OROV RNA levels increased from day 2 until day 5, which remained at higher levels on day 5 (Figure 2C). By six d.p.i., we observed a decrease in viral RNA levels and, even though by day 6 there was already high lethality, the OROV RNA levels remained relatively high in the few animals that survived to be sampled (Figure 2C). We next examined the time course of OROV replication by plaque assay. We were able to detect infectious viruses from day 2 until day 6 (Figure 2D). Similarly to RNA levels, we found an increase in the amount of infectious OROV from day 2 until day 4, which also remained high until day 5 (Figure 2D), where it reached almost 10^5^ p.f.u. mL^−1^ (day 5, *n* = 3, averaged 4.8 × 10^4^ p.f.u. mL^−1^ ± 0.2). Together, these results indicate that AG129 juvenile animals were highly susceptible to OROV infection, succumbing within six to seven days. Further, these results show that this murine animal model is able to produce high levels of infectious virus in the blood. Therefore, this model shall be useful not only to evaluate the mosquito permissibility for OROV infection, but also for mosquito–vertebrate transmission competence.

### 3.3. Ae. aegypti, Ae. albopictus, and Cx. quinquefasciatus Are Resistant to OROV Infection after Feeding on Infected Mice

As we observed that AG129 mice develop relatively high levels of viremia, and previous studies have shown that DENV and ZIKV can be transmitted from viremic AG129 mice to *Ae*. *aegypti* mosquitoes through blood meals, we tested whether this mosquito species is able to become infected after blood meals from OROV viremic mice (Figure 3A). When *Ae*. *Aegypti* mosquitoes were orally exposed to OROV through AG129 murine blood meals, we were not able to detect OROV RNA in the mosquito abdomen of both *Ae*. *Aegypti* strains, the BKK and BH strains (Figure 3B,C). Using OROV infected AG129 mice, we also tested the ability of OROV to orally infect in *Ae. albopictus* and *Cx. quinquefasciatus*. Like our *Ae*. *aegypti* data, we have not detected OROV RNA in any of these species (*Ae. albopictus* and *Cx. quinquefasciatus*) (Figure 3B,C). For both species, we also used populations established from field-collected specimens from Belo Horizonte (BH), Brazil. Consistently, the absence of OROV was ascertained in the mosquitoes at both time points, 7 and 14 days after oral exposure to infectious virus, through AG129 murine blood meals (Figure 3B,C, respectively). Thus, our results indicate that these three urban/peri-urban species (*Ae*. *aegypti*, *Ae. Albopictus*, and *Cx. quinquefasciatus*) are resistant to OROV oral infection after blood feeding on infected AG129 mice.

### 3.4. Susceptibility of Ae. aegypti, Ae. albopictus, and Cx. quinquefasciatus Mosquitoes to OROV Systemic Infection

We next investigated whether OROV was able to infect and replicate in urban mosquitoes once these initial midgut barriers are circumvented. To do this, we bypassed the midgut barriers by delivering the virus directly in the mosquito body cavity through intrathoracic nano-injection. We started by nano-injecting 1358 p.f.u. of OROV in the thorax of 5- to 7-day-old female mosquitoes and then we tested for the OROV RNA presence at 7 and 14 days post infection (d.p.i.) (Figure 4A). As the virus was injected into the mosquito thorax, we started by investigating the OROV presence in the thorax + head. When we tested this systemic delivery route in three different strains of *Ae. aegypti* (BH, RJ, and BKK), surprisingly, we found that OROV was able to infect and replicate in all three strains of *Ae*. *aegypti* (Figure 4B,C). We were able to detect OROV RNA in the thorax + head of all three mosquito strains (Figure 4B,C). Moreover, in all *Ae. aegypti* strains, we observed an infection rate of 100% at 7 d.p.i. that was retained until 14 d.p.i. (Figure 4B,C, respectively). Interestingly, we observed that, at earlier infection stages (7 d.p.i), viral titers were significantly different across all three strains of *Ae. aegypti* (χ2 Kruskal−Wallis (2) = 26.76, *p* =< 0.001 ε2 = 0.50, confidence interval (CI) 95% [0.31, 0.66], *n* obs = 55) (Figure 4B). However, we found no significant difference on the viral load at 14 d.p.i. (Figure 4C). Next, we tested whether *Ae. albopictus* and *Cx. quinquefasciatus* mosquitoes are also permissible to OROV infection and replication when the virus is directly injected into the body cavity. Likewise, in *Ae. aegypti*, we observed a rate of infection of 100% for both *Ae. albopictus* and *Cx. quinquefasciatus* (Figure 4B,C). This high infection rate was observed at both time points, 7 and 14 d.p.i. (Figure 4B,C). Furthermore, we found that, across all three species, viral titers at 14 d.p.i. remained relatively high (Figure 4C), suggesting that OROV is not only able to infect mosquito tissues, but also sustain viral replications levels sufficient to produce a persistent infection.

Our results so far indicate that, when systemically injected, OROV is able to infect and replicate in all three mosquito species. A key question is whether systemic injection in the thorax could lead to a dissemination of OROV to the abdomen, including the midgut (the first tissue infected after exposure to a viremic blood meal). To address this question, we analysed the abdomens from the same mosquitoes where the thorax + head was previously tested for the presence of OROV. Notably, we observed that, in all species and all strains tested, the abdomens were also infected with OROV (Figure 4D,E). As observed in the thorax + head results, we found an infection rate of 100% for *Ae. aegypti*. and *Ae. albopictus* at 7 and 14 d.p.i. (Figure 4D,E). In *Cx. Quinquefasciatus*, dissemination produced a lower infection rate (55%) in the abdomens at 7 d.p.i. when compared with the other two species (Figure 4D). Nevertheless, by day 14 post infection, the abdomen infection rate increased to 90% (Figure 4E). Concerning the viral titers in the abdomen of *Ae. aegypti,* we observed a significant variation across all three strains tested (χ2 Kruskal−Wallis (2) = 10.52, *p* = 0.005, ε2 = 0.19, CI 95% [0.03, 0.42], *n* obs = 55), suggesting a difference in the permissiveness to OROV infection in abdominal tissues (Figure 4D,E).

### 3.5. Upon Systemic Infection, Ae. aegypti, Ae. albopictus, and Cx. quinquefasciatus Mosquitoes Are Able to Transmit OROV to AG129 Mice

As our results show that direct injection of OROV in the body cavity results in a systemic infection, our ultimate goal was to evaluate whether infected mosquitoes could transmit the virus to a vertebrate host. To test that, we systemically infected mosquitoes by intrathoracically injecting OROV and, 14 days post infection, these mosquitoes were allowed to feed on three-week-old AG129 mice (Figure 5A). For each AG129 mouse, a maximum of five infected mosquitoes were allowed to take blood meals. Three to four days after the mosquito blood meals, mice were anaesthetised for blood collection to test for the presence of OROV RNA (Figure 5A). Using RT-qPCR to quantify the OROV RNA, we found that *Ae. aegypti* were able to transmit OROV to juvenile AG129 mice (Figure 5B). This result was observed in both strains of *Ae. aegypti* (for the BH strain, 100% of the mice became infected, whereas the transmission rate for the BKK strain was 50%) (Figure 5B).

Although, to a lesser extent, *Ae. albopictus* mosquitoes were also able to transmit OROV to AG129 mice (Figure 5B). The 25% of transmission rate observed could show that, although being a close species to *Ae. aegypti*, *Ae. albopictus* is less competent to transmit OROV after a systemic infection (Figure 5B). We then evaluated whether *Cx. quinquefasciatus* mosquitoes were also able to transmit OROV. Despite the small sample size tested, we found that, when we bypassed the midgut barrier by intrathoracic injection of the virus, *Cx quinquefasciatus* mosquitoes were also able to transmit OROV to AG129 mice (Figure 5B). Taken together, these results provide strong evidence that, once the Oropouche virus is able to reach the body cavity of *Ae. aegypti*, *Ae. albopictus*, and *Cx quinquefasciatus* mosquitoes, all these vector species are able to support viral replication and, notably, to successfully transmit the virus to a vertebrate host.

## 4. Discussion

Since OROV was first isolated in 1955 from the blood of a forest worker in Vega de Oropouche, several outbreaks involving this arbovirus have been causing urban epidemics involving thousands of people throughout tropical and subtropical regions [27,31,33]. Fortunately, OROV has not yet established a permanent urban transmission cycle. However, the history of urbanization of other arboviruses, such as CHIKV and ZIKV, suggests that viruses that are able to use humans as amplification hosts probably represent a greater risk for the establishment of permanent urban transmission. Several lines of evidence indicate that human amplification and interhuman transmission have contributed to OROV epidemics, suggesting that this arbovirus has the potential to establish permanent urban cycles [27,53,54]. Another essential requisite for an urban cycle to become permanent is the infection/transmission competence of peridomestic mosquitoes. Therefore, we evaluated the ability of OROV to infect, replicate, and be transmitted by *Ae. aegypti*, *Ae. albopictus*, and *Cx. quinquefasciatus*, three anthropophilic mosquito species. Here, we found that, when the midgut barriers are bypassed, OROV is able to systemically infect all three species. More importantly, we also have shown that these species were able to transmit OROV to a vertebrate host.

To systemically infect mosquito vectors, arboviruses must avert innate immune responses and evade or overcome several infection barriers, namely, the midgut infection barrier (MIB), midgut escape barrier (MEB), salivary gland infection barrier (SGIB), and salivary gland escape barrier (SGEB) [55,56]. Our oral infection tests indicate that OROV is not able to overcome the midgut infection barriers as we were unable to detect infected mosquitoes after both artificial membrane blood meal or through blood feeding on a natural model using infected mice. An MIB can result from either the virus not being able to enter the epithelial midgut cells or to enter the cells, but being able to replicate in order to spread to other cells and escape from the midgut. It is possible that these species do not have the appropriate receptors on the surface of midgut epithelial cells. Recent studies suggest that, rather than a single protein, arbovirus receptors consist of protein complexes involving several proteins and that the infection success of the epithelial midgut cells depends on the concentration of these proteins on the cell surface [55,57,58,59,60]. Thus, it is possible that the receptors are indeed present in the mosquito species tested here, but not in the concentrations required for an effective infection. Alternatively, the virus is able to enter the epithelial midgut cells, but is not able to overcome or evade the antiviral immune responses that limit the viral replication. Although we were unable to detect viral replication in mosquitoes after a viremic blood meal, we cannot exclude the possibility that the levels of replication were under our detection limit. Furthermore, the immune responses could be efficient enough to control and eventually eliminate the virus in the midgut before it escapes to secondary tissues. It is noteworthy that, after systemic infection by injection of the virus, we observed viral replication in the abdomen. It will be important to verify whether the midgut is one of the organs infected in the abdomen, which could help understand where the restriction of OROV infection happens.

Mosquito studies also suggest that the competence for midgut infection competence can also be dose-dependent [55,56,61,62]. For example, in a study with *Culex tarsalis*, Kramer and coauthors observed that the Western equine encephalomyelitis virus is able to infect the mosquito midgut, but unable to escape the midgut and disseminate to other organs only when low doses of virus had been ingested [62]. While several studies demonstrated that AG129 mice infected with DENV or ZIKV are able to transmit these viruses to *Ae. aegypti* mosquitoes through blood meals [37,38,40,51,63,64], we cannot rule out that OROV titers observed in the blood of AG129 mice, when the blood meals were taken (between 10^4^ and 10^5^ p.f.u. mL^−1^), are indeed insufficient to infect and overcome the midgut barriers. Concerning *Cx. quinquefasciatus*, previous studies, using viremic hamster, have shown that this vector can become infected with OROV after infectious blood meals [29]. Notwithstanding these findings, the observed infection rate was very low; two mosquitoes were infected in 326 mosquitoes tested. Additionally, the viral titers in the donor hamster were between 10^6.3^ and 10^9.9^ suckling mouse 50% lethal doses of OROV per ml, much higher than those reported for human patients with Oropouche fever (10^5.2^ to 1O^7.3^ SMLD_50/mL_) [65]. Nonetheless, we cannot discard the possibility that AG129 mice are not the appropriate model, affecting the assessment of OROV infectivity competence in the mosquito species we tested here.

To test whether viral titers in AG129 murine model were a limiting factor for the vector competence, we also tested oral infection using artificial blood meals containing 6.7 × 10^6^ p.f.u. mL^−1^ of OROV. Even using this higher dose, we were not able to detect OROV-infected mosquitoes in any of the three species. Although even higher doses of OROV could be tested using artificial blood feeding, our results presented here indicate that the refractoriness of midgut epithelial cells to OROV infection is not dose-dependent. Overall, we cannot exclude the possibility that our OROV oral infection attempts under laboratory conditions do not reflect the field complexity and geographic heterogeneity of *Cx. quinquefasciatus* population genetics. For example, a recent study demonstrated that *Cx. quinquefasciatus* mosquitoes from a colony originally established from wild-caught individuals in Florida were able to become infected and transmit OROV, although at low rates (ranging from 9.71% to 19.3% for infection; and from 0.97% to 0.82% for transmission) [66]. One possible explanation for the observed differences between these two studies is the fact that different populations were used. Another reason could be the fact that the study performed by McGregor and coauthors used a different OROV strain, a 1955 viral isolate from a febril patient in Trinidad e Tobago, whereas we used the BeAn19991 isolated from a three-toed sloth, *Bradypus tridactylus*, in the Amazon region.

The history of arboviral disease emergence shows that vector-adaptive mutations by arbovirus can increase their competence to infect and replicate in urban/peri-urban mosquitoes. This vector-adaptive evolution can be exemplified by the adaptation of CHIKV to *Ae. albopictus* mosquitoes, where envelope glycoprotein adaptive mutations acted to enhance the efficiency of entry into midgut cells to initiate infection [26,67,68,69,70,71]. A virus’s ability to successfully infect a novel mosquito vector will depend on its ability to bind to mosquito cells, replicate in suitable tissues, and avoid or suppress the mosquito immune response. In an attempt to test whether OROV is able to disseminate, infect, and replicate in secondary tissues, once the midgut barriers were circumvented, we systemically infect the virus into the mosquito body cavity. We observed surprisingly high infection rates across all three species. Our results show that OROV is able to infect and replicate in tissues from both head/thorax and from the abdomen. This in turn means that all three mosquitoes vectors species—*Ae. aegypti*, *Ae. albopictus,* and *Cx. quinquefasciatus*—have the receptors necessary for OROV to bind to mosquito cells and be internalized efficiently. It also means that OROV is capable of undergoing a productive replication in several tissues, suggesting that this arbovirus is able to avoid or successfully control the immune responses of the three tested species.

Because arbovirus transmission is associated with the ability to infect secondary tissues, namely the salivary glands, we reasoned that the observed OROV replication throughout the mosquito body could culminate in an effective transmission to a vertebrate host. Indeed, we observed high transmission rates using the AG129 murine model. Our findings suggest that, in all urban/peri-urban vectors investigated in this study, OROV infects their salivary glands and release infectious saliva, promoting AG129 mice infection after probing and feeding. Although the biological transmission is the most common among mosquito vectors, we cannot also exclude the possibility of OROV mechanical transmission (a simple transfer of viruses via mouthparts of a vector mosquito from an infected host to a susceptible one). Notwithstanding, previous mosquito studies suggested that the number of virions detected on mouthparts is not sufficient to induce infection in a naive host, indicating that mechanical transmission does not impact in the epidemiology of arbovirus such as OROV and ZIKV [29,72].

In conclusion, while this study suggests that OROV is restricted by the midgut barriers of *Ae. aegypti*, *Ae. albopictus,* and *Cx. quinquefasciatus*, it indicates that, once these barriers are avoided or suppressed, OROV has the ability to disseminate and replicate in the secondary tissues and, more importantly, be transmitted to a vertebrate host. Although it is very difficult to predict whether OROV will gain adaptive mutations that could result in efficient midgut infection competence and when this will occur, our study suggests that, if it happens, OROV can be transmitted by the three most common urban/peri-urban vector species in the Americas (*Ae. aegypti*, *Ae. albopictus*, and *Cx. quinquefasciatus*). The fact that these urban/peri-urban vector species are able to transmit OROV suggest that this arbovirus has the potential not only to establish permanent urban cycles throughout Americas, but also to spread to new geographic regions such as the continents of Africa or Asia, where potential vector species co-exist.

## Figures and Tables

**Figure 1 viruses-13-00755-f001:**
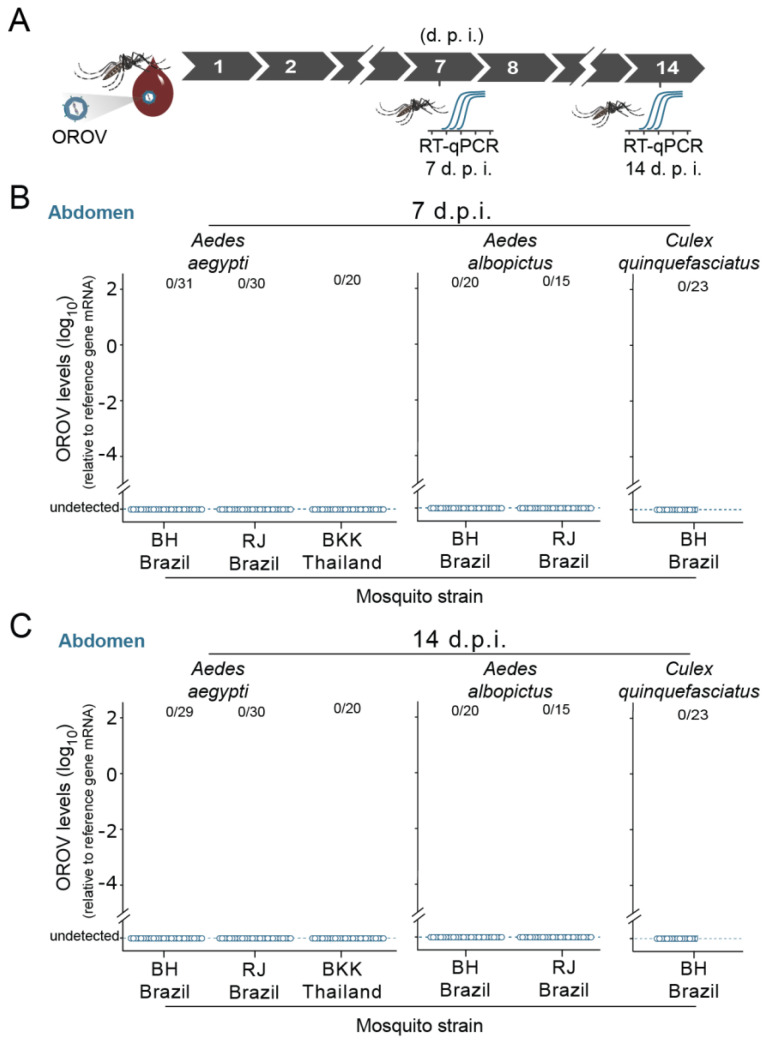
Mosquitoes *Ae. aegypti, Ae. albopictus,* and *Cx. quinquefasciatus* are resistant to *Oropouche orthobunyavirus* (OROV) oral infection after an artificial blood meal. (**A**) Scheme of the experimental design using membrane blood-feeding system to orally infect mosquitoes. Mosquitoes were allowed to take blood meals in the membrane blood-feeding apparatus containing OROV-infected blood. Seven and 14 days after the blood meal, mosquitoes were collected and tested individually for the presence of OROV. (**B**) Seven days post feeding (d.p.f.) OROV RNA levels of mosquito abdomen on blood meal containing 6.7 × 10^7^ p.f.u. mL^−1^ of OROV. RNA levels were quantified by quantitative real-time PCR (RT-qPCR). Three strains of *Aedes aegypti* were tested; that is, the BH strain; the RJ strain wild-caught population from Rio de Janeiro, Brazil); and the BKK strain. Two strains of *Aedes albopictus* were tested, the BH strain and the RJ strain (wild-caught population from Rio de Janeiro, Brazil). One strain of *Culex quinquefasciatus* was tested, the BH strain. (**C**) Fourteen d.p.f. OROV RNA levels of mosquito abdomen on blood meal containing 6.7 × 10^7^ p.f.u. mL^−1^ of OROV. Each dot represents a sample (abdomen) from an individual mosquito.

**Figure 2 viruses-13-00755-f002:**
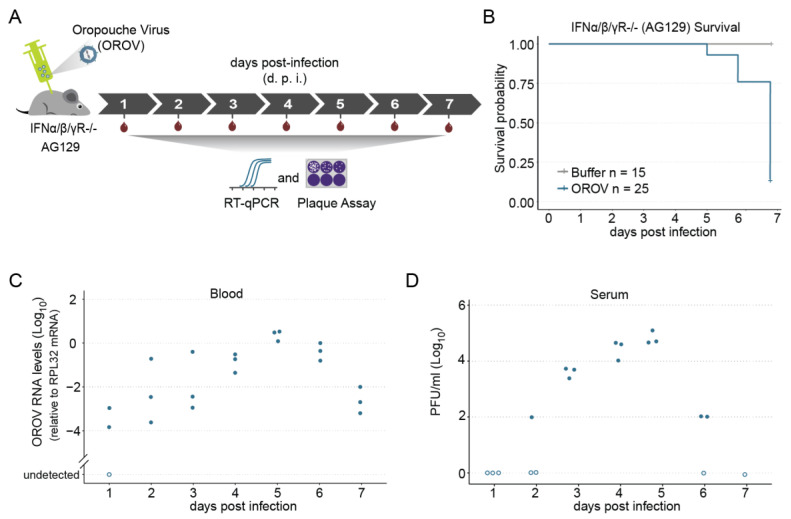
Immunodeficient AG129 (IFNα/β/γR^−/−)^ mouse model of OROV virus infection. (**A**) Scheme of the experimental design. Four- to five-week-old AG129 mice were inoculated with 10^5^ p.f.u. of OROV by intraperitoneal (IP) injection performed in the lower right quadrant. Blood was collected every 24 h during seven days for viral RNA quantification and plasma viremia titration. (**B**) Survival probability (Kaplan–Meier plot) of AG129 mice inoculated with mock or with 10^5^ p.f.u. of OROV. (**C**) RNA levels of blood samples from AG129 mice inoculated with 10^5^ p.f.u. of OROV. Samples were tested individually by RT-qPCR. Each dot represents a blood sample from an individual mouse. For each time point, three different mice were sampled. Mice were euthanized after a single blood collection. (**D**) Serum OROV titers. Serum samples were obtained from blood collected from OROV infected AG129 mice (10^5^ p.f.u. per mouse) and virus titers were measured by plaque assay.

**Figure 3 viruses-13-00755-f003:**
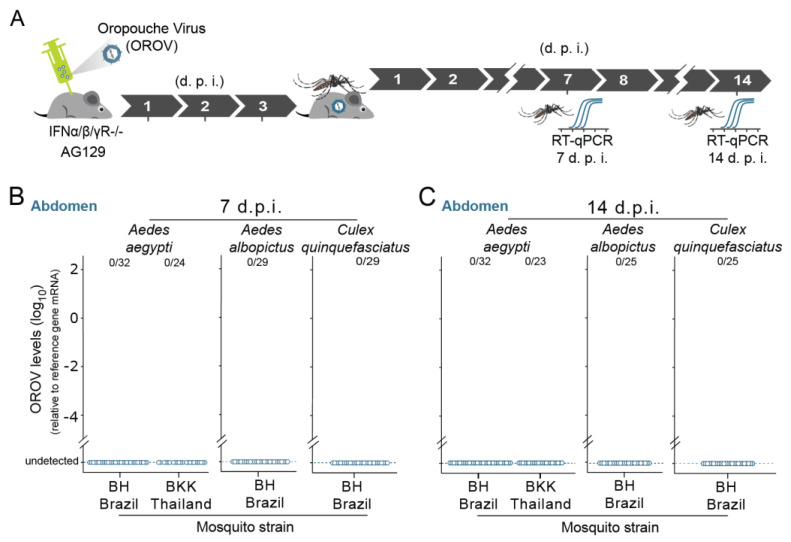
*Ae. aegypti, Ae. Albopictus,* and *Cx. quinquefasciatus* mosquitoes are resistant to OROV infection after feeding on infected mice. (**A**) Scheme of the experimental design using OROV viremic mice to orally infect mosquitoes. Four- to five-week-old AG129 mice were inoculated with 10^5^ p.f.u. of OROV by intraperitoneal (IP) injection. After three days, mice were anaesthetized and then mosquitoes (5- to 7-day-old females) were allowed to take blood meals in the OROV-infected mice. Seven and 14 days after the blood meal, mosquitoes were collected and tested individually for the presence of OROV. (**B**) Seven d.p.f OROV RNA levels of mosquito abdomen that fed on OROV-infected mice. RNA levels were quantified by RT-qPCR. Two strains of *Aedes aegypti* were tested, the BH strain (wild-caught population from Belo Horizonte, Brazil) and the BKK strain (laboratory Bangkok strain). One strain of *Aedes albopictus* was tested, the BH strain (wild-caught population from Belo Horizonte, Brazil). One strain of *Culex quinquefasciatus* was tested, the BH strain (wild-caught population from Belo Horizonte, Brazil). (**C**) Fourteen d.p.f OROV RNA levels of mosquito abdomen that were fed on OROV-infected mice.

**Figure 4 viruses-13-00755-f004:**
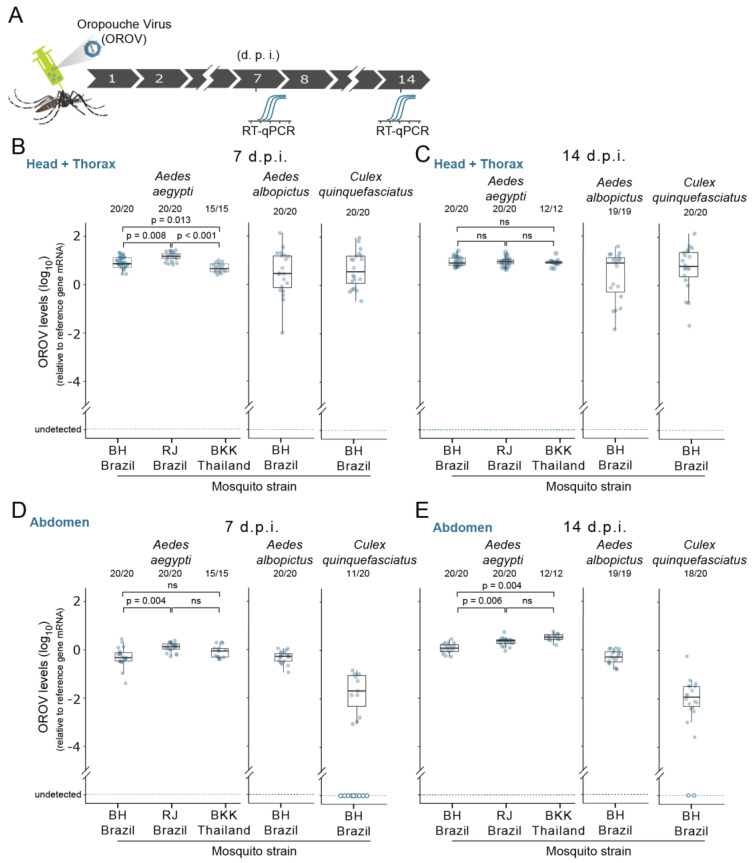
Susceptibility of *Aedes aegypti*, *Aedes albopictus*, and *Culex quinquefasciatus* mosquitoes to OROV systemic infection. (**A**) Scheme of the experimental design using intrathoracic injection to systemically infect mosquitoes. 1358 p.f.u. of OROV was injected in each mosquito; 7 and 14 days later, the head + thorax and abdomen were tested for the presence of viral RNA. (**B**) Seven d.p.i. OROV RNA levels of head + thorax from female mosquitoes injected with OROV. RNA levels were quantified by RT-qPCR. Three strains of *Aedes aegypti* were tested, BH, RJ, and BKK. One strain of *Aedes albopictus* was tested, the BH strain. One strain of *Culex quinquefasciatus* was tested, the BH strain. Each dot represents a sample (head plus thorax) from an individual mosquito. (**C**) Fourteen d.p.i. OROV RNA levels of head + thorax from female mosquitoes injected with OROV. (**D**) Seven d.p.i OROV RNA levels female mosquitoes abdomen injected with OROV. RNA levels were quantified by RT-qPCR. Three strains of *Aedes aegypti* were tested, BH, RJ, and BKK. One strain of *Aedes albopictus* was tested, the BH strain. One strain of *Culex quinquefasciatus* was tested, the BH strain. (**E**) Fourteen d.p.f OROV RNA levels female mosquitoes abdomen injected with OROV. Each dot represents a sample (abdomen) from an individual mosquito.

**Figure 5 viruses-13-00755-f005:**
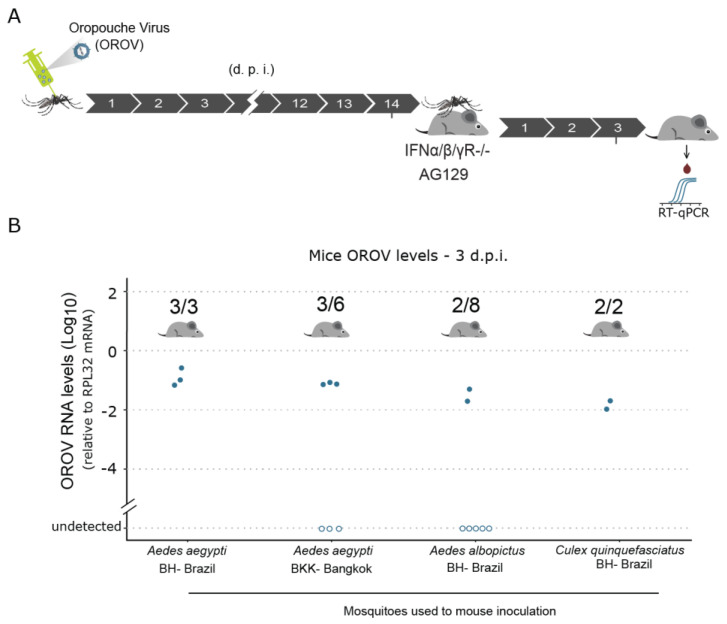
Upon systemic infection, *Ae. aegypti*, *Ae. albopictus,* and *Cx. quinquefasciatus* are able to transmit OROV to AG129 mice. (**A**) Scheme of the experimental design to test OROV transmission from mosquitoes to vertebrate host. 1358 p.f.u. of OROV was injected in each mosquito. Fourteen days later, the mosquitoes were exposed to anesthetized naïve AG129 mice for blood feeding. Three days later, the presence of OROV in the mice serum was tested by RT-qPCR. (**B**) OROV RNA levels in the mice serum three days after exposure to OROV infected mosquitoes. Mice serum samples were tested individually by RT-qPCR for the presence of OROV. Each AG129 mice was exposed to five infected mosquitoes for one hour. Between two and five mosquitoes were able to accomplish a blood meal. Full-engorged mosquitoes were counted and collected to confirm the presence of OROV by RT-qPCR. Samples were tested individually by RT-qPCR. Each dot represents a blood sample from an individual mouse.

## Data Availability

The data presented in this study are openly available in FigShare at https://doi.org/10.6084/m9.figshare.14384522.v1 (accessed on 17 March 2021).

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
