# Peer review of "Evaluation of Aedes aegypti, Aedes albopictus, and Culex quinquefasciatus Mosquitoes Competence to Oropouche virus Infection"

_viruses, 2021, doi:10.3390/v13050755_

Round 1

Reviewer 1 Report

The authors report an investigation on the probable replication of Oropouche virus (OROV)  in the anthropophilic vector mosquitoes, Aedes aegypti, Aedes albopictus and Culex quinquefasciatus. OROV is an agent of virally-induced febrile disease resembling Dengue indigenous to Central and South America, with previously-documented human case clusters. Therefore, the current work is important to assess the potential of OROV for urbanization. Briefly, the authors evaluate virus replication in mosquitoes following artificial blood meal, direct inoculation and host-feeding from AG129 mice, using standart mosquito and virus strains. Viral genome quantitation and plague assays were performed in each scenario. The experimental design is well-designed and presented clearly. The manuscript is well-written with a concise and informative background, and insights for the impact of the findings. This reviewer has no further comments, except for a particular paragraph repeated in the text (lines 470-480), that should be deleted.    

Author Response

Belo Horizonte, March 31st, 2021

To: Viruses

Ref.: Response to Reviewers (Manuscript ID: viruses-1166715)

Dear Editor,

Firstly, I would like to thank the Editor and the reviewers for your time and the comprehensive review given to our manuscript. We are glad that we had very positive reviews. Below you find point-by-point response for all raised questions and concerns. We believe now the manuscript is ready for acceptance.

Reviewer 1 comments

The authors report an investigation on the probable replication of Oropouche virus (OROV)  in the anthropophilic vector mosquitoes, Aedes aegypti, Aedes albopictus and Culex quinquefasciatus. OROV is an agent of virally-induced febrile disease resembling Dengue indigenous to Central and South America, with previously-documented human case clusters. Therefore, the current work is important to assess the potential of OROV for urbanization. Briefly, the authors evaluate virus replication in mosquitoes following artificial blood meal, direct inoculation and host-feeding from AG129 mice, using standart mosquito and virus strains. Viral genome quantification and plague assays were performed in each scenario. The experimental design is well-designed and presented clearly. The manuscript is well-written with a concise and informative background, and insights for the impact of the findings.

Point 1

This reviewer has no further comments, except for a particular paragraph repeated in the text (lines 470-480), that should be deleted.

Answer to Point 1

We appreciate the Reviewer’s input to review the revised version and give this positive comment.

We do apologize for the repeated paragraph. This was addressed in the manuscript now.

Reviewer 2 Report

The vector competence and transmission study was conducted thoroughly. I have few minor corrections or suggestions.

I would suggest to keep the title simple e.g, Evalution of Aedes aegypti, …, mosquitoes competence to Oropouche virus and transmission 

Did you evaluate for mechanical transmission? or you can briefly discuss about it?

In your findings, none of the tested mosquitoes were susceptible to OROV infection after oral feeding. However, McGregor et al, 2021 found some of  Culex arsalis and Cx. quniquefaciatus mosquitoes were infected with OROV after oral feeding and Cx. quniquefaciatus also transmitted OROV. Will you please include a brief discussion about this difference in findings between your research and other.

Author Response

Belo Horizonte, March 31st, 2021

To: Viruses

 Ref.: Response to Reviewer 2 (Manuscript ID: viruses-1166715)

Dear Editor,

Firstly, I would like to thank the Editor and the reviewers for your time and the comprehensive review given to our manuscript. We are glad that we had very positive reviews. Below you find point-by-point response for all raised questions and concerns. We believe now the manuscript is ready for acceptance.

Reviewer 2 comments

The vector competence and transmission study was conducted thoroughly. I have few minor corrections or suggestions.

Point 1

I would suggest to keep the title simple e.g, Evaluation of Aedes aegypti, …, mosquitoes competence to Oropouche virus and transmission 

Answer to Point 1

We appreciate the Reviewer’s input to review the revised version and give this positive comment. Accounting for the given suggestion, we have now change the title to: Evaluation of Aedes aegypti, Aedes albopictus and Culex quinquefasciatus mosquitoes competence to Oropouche virus infection and transmission.

Point 2

Did you evaluate for mechanical transmission? or you can briefly discuss about it?

Answer to Point 2

We acknowledge this fact pointed out by the Reviewer, we did not evaluate mechanical transmission, but as suggested by the Reviewer  we added a brief discussion exploring the subject (Please see lines 580 to 586).

Point 3

In your findings, none of the tested mosquitoes were susceptible to OROV infection after oral feeding. However, McGregor et al, 2021 found some of  Culex arsalis and Cx. quniquefaciatus mosquitoes were infected with OROV after oral feeding and Cx. quniquefaciatus also transmitted OROV. Will you please include a brief discussion about this difference in findings between your research and other.

Answer to Point 3

We acknowledge this fact pointed out by the Reviwer, we developed a  brief discussion exploring the differences between the two studies that could explain the findings heterogeneousness (please seen lines 546 to 555).